# Exploring Immunological Effects and Novel Immune Adjuvants in Immunotherapy for Salivary Gland Cancers

**DOI:** 10.3390/cancers16061205

**Published:** 2024-03-19

**Authors:** Ryosuke Sato, Hidekiyo Yamaki, Hiroki Komatsuda, Risa Wakisaka, Takahiro Inoue, Takumi Kumai, Miki Takahara

**Affiliations:** 1Department of Otolaryngology-Head and Neck Surgery, Asahikawa Medical University, Asahikawa 0788510, Japan; rsato@asahikawa-med.ac.jp (R.S.); hidekiyo@asahikawa-med.ac.jp (H.Y.); komatsuda@asahikawa-med.ac.jp (H.K.); r-wakisaka@asahikawa-med.ac.jp (R.W.); takapiro9242@gmail.com (T.I.); miki@asahikawa-med.ac.jp (M.T.); 2Department of Innovative Head & Neck Cancer Research and Treatment, Asahikawa Medical University, Asahikawa 0788510, Japan

**Keywords:** salivary gland cancer, immunotherapy, tumor microenvironment, immune checkpoint molecule, programmed death ligand-1

## Abstract

**Simple Summary:**

The efficacy of immunotherapies in salivary gland cancer (SGC) remains controversial. To optimize immunotherapy, understanding the tumor microenvironment (TME) of SGC is necessary. In this review, we demonstrate that high-grade mucoepidermoid carcinoma and salivary duct carcinoma exhibit immune-hot TME. In contrast, adenoid cystic carcinomas exhibit an immune-cold TME. While the reported efficacy of immune checkpoint inhibitors (ICIs) for SGCs is generally poor, several studies have shown promising clinical efficacy of ICIs indicating that ICIs might be beneficial for a specific population of SGC. Molecule-targeted therapies have shown promising clinical efficacy against SGC. Recent evidence indicates that these molecules could be targets for antigen-specific immunotherapies. This review discusses the current understanding and future directions of immunotherapies for SGCs.

**Abstract:**

Salivary gland cancer (SGC) is rare and comprises over 20 histological subtypes. Recently, clinical experience regarding immunotherapies for SGCs has been accumulating, yet their efficacy remains controversial. Understanding the tumor microenvironment (TME), including the expression of immune checkpoint molecules in SGC, is crucial to optimizing immunotherapy. In this review, we demonstrate that high-grade mucoepidermoid carcinoma and salivary duct carcinoma generally exhibit immune-hot TME with high immune cell infiltration, frequent genetic mutations, and robust immune checkpoint molecule expression. In contrast, adenoid cystic carcinomas exhibit an immune-cold TME. While the reported efficacy of immune checkpoint inhibitors (ICIs) for SGCs is generally poor, several studies showed promising clinical efficacy of ICIs, with an objective response rate ranging from 20.0–33.3%, indicating that ICIs might be beneficial for a specific population of SGC. Molecule-targeted therapies including anti-human epidermal growth factor receptor 2 and anti-androgen receptor therapies have shown promising clinical efficacy against SGC. Recent evidence indicates that these molecules could be targets for antigen-specific immunotherapies including chimeric antigen receptor-T therapy and cancer vaccines. This review discusses the current understanding and future directions of immunotherapies for SGCs, including ongoing clinical trials.

## 1. Introduction

Salivary gland cancer (SGC) is a rare type of cancer (0.6–1.4 per 100,000), accounting for less than 3% of all head and neck cancers [1]. According to the WHO classification of head and neck cancers in 2017 [2], SGCs originate from major or minor salivary glands and encompass more than 20 histological subtypes. Surgery remains the gold standard treatment for resectable SGCs, often followed by adjuvant radiotherapy for patients with positive surgical margins, lymph node metastases, or high-grade tumors. However, due to its rarity and diverse histology, there is no clinically established therapy for inoperable tumors, invasive local recurrences, or distant metastases [3]. Systemic chemotherapies have been investigated for the treatment of unresectable SGCs. Monotherapy with cytotoxic anticancer drugs such as cisplatin, paclitaxel, and vinorelbine has not demonstrated satisfactory clinical efficacy [4,5,6]. However, combinations of cytotoxic anticancer drugs have shown moderate clinical efficacy, with an objective response rate (ORR) ranging from 15% to 70% [7]. Recently, attention has turned to molecular-targeted therapies in light of advancements in the molecular biology of cancers. In salivary duct carcinoma (SDC), therapies targeting human epidermal growth factor receptor 2 (HER2) and androgen receptor (AR) have shown promising clinical efficacy [8]. Systemic chemotherapies, including molecular-targeted therapies, are generally used in patients with metastatic or recurrent SGCs [7], but their effectiveness as a first-line treatment remains to be elucidated.

Immunotherapy is a novel cancer therapy that harnesses the immune system of the body. Immune cells play an important role in recognizing and eliminating tumor cells based on tumor antigens. However, tumor cells can evade the immune system through various mechanisms, including the activation of negative immune checkpoints. Immune checkpoints are immunosuppressive molecules expressed on tumor and suppressive immune cells, enabling tumor immune evasion. Immune checkpoint inhibitors (ICIs) activate the self-immune system by blocking negative immune checkpoints, such as programmed death receptor-1 (PD-1) and cytotoxic T-lymphocyte-associated protein 4 (CTLA-4) [9]. ICIs have been approved for multiple types of cancers extending the survival of patients with these cancers [10]. Recently, clinical experience with immunotherapies in SGCs has been accumulating. Some reports have shown promising clinical efficacy [11,12] while others have reported no response to ICIs in SGCs [13]. This discrepancy may be mediated by differences in the tumor microenvironment (TME) across histological types of SGCs. The TME, comprising tumor cells, immune cells, and cytokines, affects antitumor immunity and is pivotal in determining the clinical efficacy of immunotherapies [3]. Thus, understanding the TME differences among histological types of SGCs is important for enhancing immunotherapies in SGCs. Theocharis et al. [3] and Mueller et al. [8] have summarized the immune regulatory cells and immune checkpoint expression in SGC overall. This scoping review discusses unique topics including differences in the immunological backgrounds of representative histological types of SGCs, including mucoepidermoid carcinoma (MEC), adenoid cystic carcinoma (AdCC), and SDC. We also explore current evidence and future perspectives regarding checkpoint blockade and antigen-specific immunotherapies in SGCs.

## 2. Methods

This review referred to the Preferred Reporting Items for Systematic reviews and Meta-Analyses (PRISMA) extension for scoping reviews [14]. The existing literature about the immunological background and the immunotherapies of SGCs was reviewed through PubMed and Google Scholar until January 2024. Two authors (R.S. and H.Y.) reviewed a selection of the literature. The search strategy was carried out by various combinations of terms: “salivary gland cancer”, “mucoepidermoid carcinoma”, “salivary duct carcinoma”, “adenoid cystic carcinoma”, “tumor microenvironment”, “immunotherapy”, “immune checkpoint”, and “programmed death ligand-1”. The review included not only clinical research but also an internal analysis of clinical trials, basic research, and case reports if the literature gave relevant information for the review. The literature not written in English was excluded.

## 3. The Immunological Background of Mucoepidermoid Carcinoma

MECs represent the most common histological subtype comprising 35% of the SGCs. MECs are classified as low-, intermediate-, or high-grade based on their histological type [15]. Although the prognosis of low-grade MEC is generally favorable after surgical resection, the 5-year survival rate of high-grade MEC is low, ranging from 0–43%, even with the multimodal treatment [16]. Suppression of antitumor immunity contributes to MEC progression compared to benign salivary gland tumors; MECs exhibit higher expression levels of several immune checkpoint molecules, including lymphocyte activation gene 3 (LAG3), T-cell immunoglobulin, mucin domain-containing protein 3 (TIM3), and adenosine 2a receptor [17]. Knockdown of these immune checkpoint molecules prevents tumor progression in SGC mouse models, indicating that immune checkpoint molecules may support MEC growth. Abnormal angiogenesis by vascular endothelial growth factor (VEGF) in the TME initiates tumor formation and progression, partly via immune modulation [18]. VEGF is expressed in MEC regardless of histological grade, and inhibition of VEGF decreases tube formation in MEC cell lines [19]. VEGF-A expression correlates with the presence of tumor-associated macrophages (TAM) in patients with MEC [19]. As TAMs promote the migration and invasion ability of MEC cells [20], angiogenesis and TAMs cooperate to create an immunosuppressive TME conducive to MEC progression.

The immunogenicity of the TME in MEC varies across histological grades. The expression of programmed death ligand-1 (PD-L1) and human leukocyte antigen (HLA) positively correlates with histological grade [21]. Additionally, high-grade MEC exhibits a higher tumor mutational burden (TMB) than low-grade MEC [22]. Using RNA-sequencing analysis, Kang et al. reported that MEC with immune-hot TME exhibit activated immunity characterized by T-cell infiltration, cytolytic score, interferon-γ, antigen-presenting machinery, and immune modulator genes. Immune checkpoint molecules, including PD-L1, programmed death ligand-2 (PD-L2), T-cell immunoreceptors with immunoglobulin and immunoreceptor tyrosine-based inhibitory motif domains, and CTLA-4, are abundantly expressed in MECs with immune-hot TME compared to those with immune-cold TME [23]. Among SGCs, MECs show higher immune cell infiltration in the TME, greater T-cell receptor diversity, and increased expression of HLA class I and II than that of AdCCs [24]. Given that MEC may represent an immunogenetic tumor type where immunotherapies could be effective (Figure 1), further research is necessary to examine the immunological differences in the TME across different histological grades of MEC.

## 4. The Immunological Background of Salivary Duct Carcinoma

SDC represents one of the most aggressive subtypes of SGCs, accounting for approximately 10% of cases, with high rates of local recurrence and distant metastasis [25,26]. About 20–59% of SDC arise from pleomorphic adenoma [26]. The prognosis of SDC is poor, with 5-year disease-free survival (DFS) and overall survival (OS) rates in the range of 42.5–52.7% and 34.1–41%, respectively [27,28]. SDC is considered a relatively immunogenic tumor with elevated immune cell infiltration and gene mutations. The immune-infiltrated subtype constitutes 64% of SDC cases [29]. RNA-sequencing studies have revealed increased T-cell infiltration and neoantigen expression in SDCs compared to that of myoepithelial carcinoma or AdCC [30]. In addition to immune cell infiltration, another study has shown that SDCs express high levels of T-cell receptor diversity and HLA class I/II [24]. However, multiple immunosuppressive mechanisms co-exist in the TME of SDCs. Despite upregulated CD8+ T-cell infiltration, the expression of immune checkpoint molecules, including FOXP3, PD-1, CTLA-4, and LAG-3, is associated with aggressive histological features of SDCs [28]. In a study involving 175 patients with SDC, Hirai et al. reported that 18% of patients were classified as PD-L1 positive (tumor proportion score ≥ 1%), and the positive PD-L1 expression was associated with worse progression-free survival (PFS) and OS in multivariate analysis [28]. Another study observed HLA class I downregulation in 82% of SDCs [31]. A high proportion of M2 TAM, known to suppress antitumor immunity, was observed in both the immune-hot and immune-cold TME of SDCs [29]. These TAMs expressed immune checkpoint molecules, including PD-L1, CD86, TIM-3, and galectin-9. Collectively, SDC represents an immune-hot tumor, but the coexistence of immunosuppressive TMEs may contribute to immune cell exhaustion and SDC progression.

## 5. The Immunological Background of Adenoid Cystic Carcinoma

AdCC accounts for approximately 10% of SGCs characterized by a slow growth but a high rate of distant metastasis [32,33]. Although the survival of patients with bone metastasis is poor, many patients with recurrent or metastatic AdCC, especially with pulmonary metastasis, survive for a long time due to an indolent clinical course. While radiation therapy (RT) is used for the targeted or palliative treatment of distant metastases, a large retrospective analysis indicated that radiotherapy does not improve OS [34]. Systemic chemotherapy has also failed to provide lasting benefits [6,35]. A critical aspect of AdCC is its immune-depleted TME. AdCC displays unfavorable immunogenic profiles for immunotherapies, including a T-cell exclusion phenotype, low immune cell infiltration and TMB, significant infiltration of M2 TAM and myeloid-derived suppressor cells, and HLA class I loss [30]. M2 TAM promotes AdCC proliferation via chemokines. The CCL2/CCR2 axis, a chemokine that modulates the TME, is involved in the recruitment and polarization of M2 TAM, and the overexpression of CCL2 is obviously associated with the poor prognosis of AdCC [36]. AdCC exhibits a low infiltration of immune cells, such as CD8+, granzyme B-positive tumor-infiltrating lymphocytes (TILs), CD1a+, and CD83+ cell populations [37]. Reduced numbers of CD8+ and CD1a+ cells in the TME are significantly associated with high recurrence rates and short survival [37] indicating that the immune desert TME may contribute to the progression of AdCC. Additionally, the expression of immune checkpoint proteins is low in patients with AdCC. Four out of six studies have shown no tissue PD-L1 expression in AdCCs [13,21,38,39,40,41]. Compared with other histological types, LAG3 expression in TIL and TP53 mutations in tumors are low in AdCC [42].

Besides TP53, other genetic mutations such as MYB-NFIB or MYBL1-NFIB fusions are observed in approximately 60% of AdCC [43,44,45]. MYB overexpression upregulates several neovascularization factors, including VEGF-A, fibroblast growth factor 2, and KIT, in AdCC [43]. Although the clinical benefits of multi-kinase inhibitors targeting these neovascularization factors are expected, monotherapies with these inhibitors have shown unsatisfactory results in AdCC [46,47,48,49,50,51]. Since multi-kinase inhibitors are known to regulate tumor immunity in other types of cancers [52], immunomodulation of the TME by these reagents remains unresolved in AdCC. Thus, the mutated AdCC proteins may act as immunogenic antigens. Several patients with metastatic AdCC have circulating T cells that recognize peptides associated with MYB-NFIB fusion [53]. The immune-cold TME in AdCC, characterized by low immune cell infiltration, frequency of mutations, and expression of immune checkpoint molecules (Figure 1), could be overcome by activating these mutation-reactive T cells.

## 6. The Expression Rates and Prognostic Ability of Immune Checkpoint Molecules in Salivary Gland Cancer

Like SDC, immune checkpoint molecules play an important role in the progression of SGC. Among the immune checkpoints, tissue expression of PD-L1 is the most investigated molecule as a poor prognostic and predictive factor for PD-1 inhibitors in various types of cancers [54]. The expression rates of PD-L1 in representative histological types of SGCs are summarized in Table 1 [21,28,31,38,39,40,41,55,56,57,58,59,60,61,62,63,64,65,66,67,68,69,70,71,72,73]. Considering a cutoff point of ≥1% positive tumor cells, PD-L1 expression was observed in approximately 15–30% of SGCs [28,40,41,55,60,61,62,63,66,70,72,73]. Few studies showed high (>50%) [65,70] or no expression rates of PD-L1 [40]. PD-L1 positivity rates in SDC and adenocarcinoma not otherwise specified (adenocarcinoma NOS) are generally high ranging approximately 20–100% [28,31,55,56,62,67,71]. MEC showed moderately high PD-L1 positivity rates of approximately 10–80% [55,57,61,65,66,69,71,73]. The expression rate of PD-L1 in AdCC and acinic cell carcinoma was generally low [21,38,41,55,58,59,61,65,72]. Collectively, SDC, adenocarcinoma NOS, and MEC are possible candidates for treatment with PD-1 inhibitors.

In addition to other cancer types, positive PD-L1 expression may predict a poor prognosis in SGCs. Nine studies reported that positive PD-L1 expression was associated with poor prognosis in SGCs [28,31,55,58,61,64,65,69,71]. In contrast, seven studies reported no association between PD-L1 positivity and prognosis in SGCs [56,57,60,62,66,68,70]. Only one study reported a favorable prognosis for PD-L1-positive SGCs (low recurrence rate and prolonged DFS) [59]. The discrepancy between these studies might be explained by the histological types, PD-L1 scoring methods, cutoff points, and antibodies used to detect PD-L1. Further studies are required to standardize the method to detect PD-L1 expression and to apply PD-L1 as an accurate prognostic marker in SGCs. Wu et al. conducted a meta-analysis regarding the association between PD-L1 positivity and prognosis in SGCs [74]. This study revealed that high PD-L1 expression is associated with poor OS and DFS, indicating the importance of PD-L1 in SGC progression.

PD-L2, another ligand of PD-1, is also a prognostic factor in other types of cancers [75,76,77]. The positive expression rate of PD-L2 (58.7%) was higher than PD-L1 (25.4%) in SGCs [57]. Nakano et al. reported that PD-L2 expression is associated with reduced disease-specific survival and DFS in SGCs [60]. Interestingly, the expression of PD-L2 is observed in AdCC, whereas PD-L1 expression is usually negative [37,38]. Since infiltrated lymphocytes are positive for PD-1, the function of these lymphocytes may be suppressed via PD-L2 instead of PD-L1 in AdCC. Other negative immune checkpoints, including HLA-E and HLA-G, may also inhibit antitumor immunity against SGCs [21,37]. Lymphocytes express exhausted markers, such as CTLA-4 and LAG3, in the TME of SGCs [28,29,42].

## 7. The Clinical Efficacy of Immune Checkpoint Inhibitors in Salivary Gland Cancers

Being expressed in SGCs, negative immune checkpoints can be attractive targets to achieve tumor regression by releasing antitumor immunity. The clinical efficacy of ICIs in SGCs is summarized in Table 2 [11,12,13,72,78,79,80,81,82,83,84,85,86]. Generally, the clinical efficacy of ICI monotherapy is poor, with an ORR of less than 20% [13,72,78,79,80,82,83,85,86]. Although combination regimens with ICIs have been attempted in SGCs to improve their therapeutic effects, the results remain disappointing.

A combination of nivolumab and ipilimumab showed that ORR in AdCC and non-AdCC was 4% and 9%, respectively [86]. Another study with nivolumab and ipilimumab reported that the ORR of AdCC and non-AdCC was 6% and 16%, respectively [72]. Combination therapy with pembrolizumab and RT showed no objective response outside of the radiation treatment field [13]. Combining pembrolizumab with vorinostat, a histone deacetylase inhibitor, resulted in an ORR of 16% [81]. A phase II study of pembrolizumab with a vascularization inhibitor (lenvatinib) in AdCC failed to demonstrate superiority against the reported efficacy of lenvatinib monotherapy (ORR: 6%) [85]. Based on the promising ORR ranging from 16–42% in SGCs, mainly composed of SDC, by anti-androgen therapies [87,88,89], a multicenter phase II single-arm clinical trial with pembrolizumab and goserelin, which inhibit androgen production, is ongoing in the patients with AR-positive SGCs (NCT 03942653). Interim analysis of this study reported that the ORR, clinical benefit rate, and 6-month PFS were 22%, 88%, and 63%, respectively [90]. Although most ICI studies, including combination regimens, have failed to demonstrate clinical efficacy overall in SGCs, two clinical trials combining ICIs and cytotoxic chemotherapy are ongoing. NCT 03360890 is a phase II single-arm clinical trial investigating the clinical efficacy of pembrolizumab combined with docetaxel in SGCs and thyroid cancer. Another phase II single-arm clinical trial using pembrolizumab combined with pemetrexed for SGCs is ongoing (NCT 04895735).

As described in the previous section, the expression of PD-1 varies among the histological types of SGCs. We previously reported a promising ORR (33.3%) with PD-1 inhibitors in patients with high-grade SGCs, including SDC and high-grade MEC that express PD-1 [11]. Aggressive histological types of SGCs have been reported to show high-PD-L1 expression and TMB, which are positive predictive factors for the favorable clinical efficacy of PD-1 inhibitors [66,91,92]. AdCC is characterized by an immune-cold TME and negative PD-L1 expression. As expected, the ORRs for ICIs in AdCC were poor, in the range of 4.0–8.7% [72,78,85,86]. Since immune checkpoint-negative SGCs may be unresponsive to ICIs, it is rational to recruit patients with PD-L1- or PD-L2-positive SGCs for treatment with PD-1 inhibitors. Even et al. reported that the ORR for pembrolizumab was higher in PD-L1-positive (10.7%) compared to PD-L1-negative SGCs (2.6%) [83]. As several studies have reported promising ORRs of 20.0–33.3% [11,12,84], the appropriate selection of patients based on histological types and biomarkers, including PD-L1 and PD-L2, may improve the insufficient clinical efficacy of ICIs in SGCs.

## 8. Target-Specific Immunotherapy against Salivary Gland Cancers

A drawback of ICIs is their non-specific activation of immune cells. Autoimmune T cells activated by ICIs cause immune-related adverse effects. Tumor-specific immunotherapies are designed to avoid the risk of autoimmunity by targeting specific proteins expressed in tumor cells but not in normal cells [93]. To establish tumor-specific immunotherapy, the identification of tumor-derived molecules is necessary. Targeted therapies against specific molecules expressed in tumor cells have been investigated in SGCs. Inhibitors of AR, HER2, and neurotrophic tyrosine receptor kinases have shown promising clinical efficacy in patients with SDC and secretory carcinoma [94]. Among the targeted therapies, monoclonal antibodies bind to specific proteins on the tumor surface and exhibit therapeutic effects by blocking tumor growth signals. Moreover, the antitumor efficacy of monoclonal antibodies partly depends on natural killer cells, known as antibody-dependent cellular cytotoxicity [9,95]. Because the expression of HER2 is generally high in patients with SDC and adenocarcinoma NOS [59], anti-HER2 monoclonal antibody (trastuzumab) combined with docetaxel showed excellent ORR ranging from 60–72% in patients with SDC [96,97,98]. The epidermal growth factor receptor (EGFR) is expressed in various histological types of SGCs [59,94]. Despite some studies with anti-EGFR therapies not showing clinical efficacy against SGCs [99,100,101], we reported that the anti-EGFR monoclonal antibody (cetuximab) combined with paclitaxel demonstrated a preferable ORR of 71.4% in the patients with SGCs mainly composed of SDC [102].

These therapeutic antibodies can be used to establish chimeric antigen receptor (CAR) T cells, which is a mode of target-specific immunotherapy. CAR T cells are patient-derived T cells genetically transduced with Fab regions of tumor-reactive antibodies. The efficacy of CAR T-cell therapy has been demonstrated in hematologic malignancies, for which CD19-targeting CAR T-cell therapies have been approved by the U.S. Food and Drug Administration [9]. Invariant natural killer T cells transduced with mesothelin-targeting CAR are cytotoxic against mesothelin-expressing cancer cell lines, including SGC cells [103]. Prostate-specific membrane antigen (PSMA)-targeting CAR T-cell therapy decreases serum prostate-specific antigen levels in prostate cancer, indicating the feasibility of CAR T-cell therapy in solid tumors [104,105,106]. PSMA is a target of CAR T cells, antibody drug conjugates, and bi-specific T-cell-directed therapy in prostate cancer. In addition to prostate cancer, PSMA is expressed in SGCs such as AdCC. The studies on immunohistochemistry and PSMA/positron emission tomography revealed that the positivity rates of PSMA in AdCC were 94% and 93%, respectively [107,108]. The study on radioligand therapy by ^177^Lu-EB- PSMA-617 shows promising clinical response [109,110], suggesting that PSMA could be a targetable antigen in immunotherapy against AdCC. A phase I clinical trial of PSMA-targeting CAR T-cell therapy for patients with SGC or prostate cancer is ongoing (NCT 04249947). Phase I clinical trials with HER2-targeting CAR T-cell and CAR-macrophage therapies have recruited patients with HER2-positive solid tumors (NCT 04511871, 04660929, 06241456, and 06254807). As the safety of HER2-targeting CAR T-cell therapy has been confirmed [111], this treatment would be an interesting approach for SDC, most of which express HER2.

The requirements of ex vivo transduction and the expansion of T cells in the laboratory are major hurdles in applying CAR T therapy to a broad population of patients. Furthermore, the benefits of time-consuming and expensive CAR T-cell therapy can be easily impaired because of tumor heterogeneity and antigen loss [103,104]. Cancer vaccines are promising tumor-targeting immunotherapies that induce tumor-reactive T cells in patients by administering antigens and adjuvants. Because there is no need for expensive genetic transduction and T-cell expansion ex vivo, it is relatively simple to switch antigens in cancer vaccines compared to CAR T therapy, in situations of antigen loss. An enormous number of epitopes derived from tumor-derived antigens have been identified as vaccine sources [93]. After vaccine administration, these epitopes bind to HLA molecules on antigen-presenting cells, followed by the expansion of tumor-reactive T cells. Wilm’s tumor gene 1 (WT1) is a tumor-associated antigen (TAAs) that is overexpressed in tumors compared to normal tissues. Cancer vaccines with WT1-derived epitopes have been investigated in several types of cancers including SGCs [112,113,114]. In carcinoma ex pleomorphic adenoma, peptide-based WT1-targeting vaccines increased CD8+ T cells recognizing the WT1-derived peptide [113]. Another study reported that a WT1-targeting vaccine suppressed tumor growth for one year in recurrent AdCC [114].

In addition to TAAs, recent advances in genetic sequencing have enabled the detection of tumor-specific antigens (TSA) that are expressed only in tumor cells. TSA is considered an ideal cancer vaccine target because of its high immunogenicity and low cross-reactivity with normal tissues [93]. Neoantigens are TSAs generated by mutations in tumor cells. Cancer vaccines against neoantigens show promising clinical efficacy against several types of cancers [115,116,117,118]. The genetic sequences of SGCs revealed that SDC had a more abundant expression of neoantigens than other histological types of SGC, indicating that patients with SDC would benefit from cancer vaccines targeting neoantigens [30,92]. As a proof of concept, a patient with metastatic SDC achieved a complete response using a combination approach of a neoantigen-based dendritic cell vaccine and nivolumab [119]. AdCC is an unfavorable target for neoantigen-based vaccines because of its low mutational burden. Since it is difficult to conduct clinical trials in SGC owing to its rarity, clinical trials regarding neoantigen-based cancer vaccines are ongoing in solid tumors. Two clinical trials are investigating the efficacy of peptide- and mRNA-based neoantigen vaccines combined with pembrolizumab in patients with solid tumors (NCT 05198752 and 05269381). Because some SGCs overexpress neoantigens, the results of these trials would support the establishment of a TSA-targeting vaccine for SGC.

## 9. Conclusions and Future Directions

In this review, we summarize the current knowledge regarding the immune microenvironment in SGC. Although the clinical efficacy of ICI monotherapy is unacceptable, the optimization of combination therapies, such as multi-kinase inhibitors and RT, is promising for enhancing antitumor immunity by modifying the TME in SGCs. Lenvatinib is a multi-kinase inhibitor that targets VEGFR2 and VEGFR3. Interestingly, the inhibition of VEGF/VEGFR signaling reduces the number of regulatory T cells and increases that of cytotoxic T cells in hepatocellular carcinoma [120]. In a mouse model, lenvatinib with an anti-PD-1 antibody activated CD8+ T cells by reducing TAMs and activating the interferon pathway in multiple types of cancer cell lines [121]. Although the impact of multi-kinase inhibitors on the TME in SGC has not been determined, VEGF is expressed in MEC and immune-cold AdCCs [122,123] suggesting that multi-kinase inhibitors have the potential to augment the therapeutic efficacy of immunotherapy in SGCs. RT is also considered a candidate for combination therapy with immunotherapies. In addition to its direct tumor cytotoxicity through free radical-mediated DNA damage, the immunomodulatory effects of RT on the TME have been revealed in various types of cancers [124]. In AdCC, RT increased both the number of CD8+ lymphocytes and the ratio of CD8+/FoxP3+ regulatory T cells in the TME [38]. One case report described that low-dose RT combined with anti-PD-1 therapy resulted in long-standing stable disease in a patient with recurrent AdCC [125]. In contrast, another study reported the immunosuppressive effect of RT by decreasing the CD4+/regulatory T-cell ratio in AdCCs [126]. Further research is required to confirm the immunomodulatory effects of RT on the TME of SGC.

Establishing firm evidence for treatment strategies through clinical trials is difficult in rare types of cancer, including SGCs. Since the clinical evidence of chemotherapy is scarce in SGC, the combination of HER2-targeting antibody (trastuzumab) and docetaxel has shown favorable clinical responses with 70% of ORR [96]. Because trastuzumab is based on the IgG1 subclass, antitumor activity via antibody-dependent cellular cytotoxicity is expected. Docetaxel has shown synergy with a Th1-skewed cytokine [127] and a tumor antigen-based vaccine [128] in preclinical models. In a phase II trial of prostate cancer, docetaxel showed a synergistic effect with a cancer vaccine [129]. Thus, the combination of trastuzumab and docetaxel, which has been clinically approved as an efficient chemotherapy for SGC, may be an interesting adjuvant for immunotherapy. Although many clinical trials of cancer vaccines have been conducted, their outcomes have been unfavorable [93]. These poor results can be partly explained by the inadequate selection of adjuvants. To improve the effects of immunotherapy in SGC, further research is necessary to examine the immunologic effects of current treatments, identify novel immune adjuvants, such as pattern recognition receptor ligands [130], and establish cell lines from each histological subtype of SGCs.

## Figures and Tables

**Figure 1 cancers-16-01205-f001:**
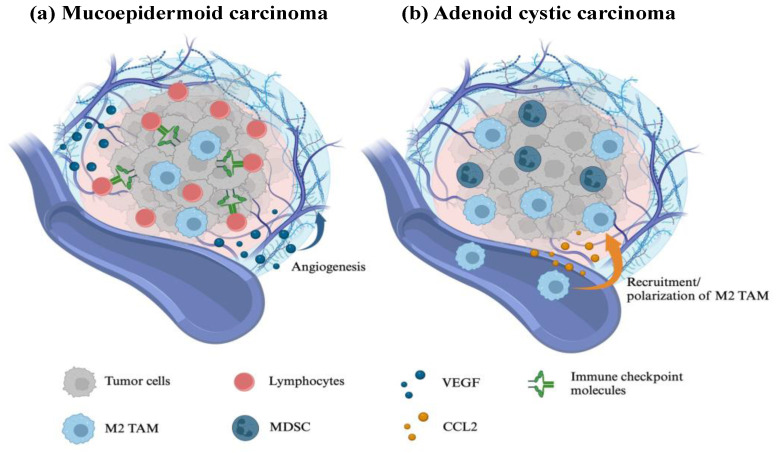
Comparison of tumor microenvironment between (**a**) mucoepidermoid carcinoma and (**b**) adenoid cystic carcinoma. Infiltration of immune cells and expression of immune checkpoint molecules are relatively high in mucoepidermoid carcinomas. Immune regulatory cells, including MDSC, infiltrate the tumor microenvironment of adenoid cystic carcinoma. MDSC, myeloid-derived suppressor cells; TAMs, tumor-associated macrophages; VEGF, vascular endothelial growth factor.

**Table 1 cancers-16-01205-t001:** Expression rates of PD-L1 in major histological types of salivary gland cancers.

No.	Author	PD-L1 Scoring	Cut Off	SDC	Adenocarcinoma-NOS	MEC	AdCC	ACC	Others	Total
1	Mukaigawa et al., 2016 [55]	TC	≥1%	15/31 (48%)	4/11 (36%)	3/34 (9%)	1/53 (2%)	0/18 (0%)	9/28 (32%)	50/219 (23%)
		IC	≥1%	9/31 (29%)	2/11 (18%)	2/34 (6%)	0/53 (0%)	1/18 (6%)	6/28 (21%)	28/219 (13%)
2	Sridharan et al., 2016 [38]	TC	≥5%				0/21 (0%)			0/21 (0%)
3	Haderlein et al., 2017 [56]	CPS	≥5	11/50 (22%)						11/50 (22%)
4	Chang et al., 2017 [57]	H-score	≥1	5/11 (45%)		7/27 (26%)	4/15 (27%)			16/53 (30%)
5	Harada et al., 2018 [58]	TC	≥5%		5/9 (56%)	7/11 (64%)	11/25 (44%)		1/2 (50%)	24/47 (51%)
6	Szewczyk et al., 2019 [59]	TC	≥5%	1/16 (6%)	1/13 (8%)	3/16 (19%)	10/33 (30%)	0/15 (0%)	0/9 (0%)	20/115 (17%)
		IC	≥5%	2/16 (13%)	0/13 (0%)	1/16 (6%)	0/33 (0%)	0/15 (0%)	0/9 (0%)	3/115 (3%)
7	Nakano et al., 2019 [60]	TC	≥1%	4/8 (50%)		4/7 (57%)	0/11 (0%)		1/4 (25%)	11/32 (34%)
8	Vital et al., 2019 [61]	TC	≥1%	3/10 (30%)	2/12 (17%)	9/36 (25%)	3/36 (8%)		7/37 (19%)	28/161 (17%)
9	Mosconi et al., 2019 [37]	TC	≥10%				0/36 (0%)			0/36 (0%)
10	Hamza et al., 2019 [62]	TC	≥1%	29/113 (26%)						29/113 (26%)
11	Gargano et al., 2019 [63]	TC	≥1%	5/28 (18%)						5/28 (18%)
12	Xu et al., 2019 [31]	CPS	≥1	20/36 (56%)					15/17 (88%)	35/53 (66%)
13	Kesar et al., 2020 [64]	TC	≥5%		6/26 (23%)	2/10 (20%)	0/16 (0%)	1/7 (14%)	4/8 (50%)	13/67 (19%)
14	Witte et al., 2020 [65]	TC	≥1%	1/1 (100%)	12/12 (100%)	11/21 (52%)	16/41 (39%)	13/16 (81%)	3/3 (100%)	61/94 (65%)
		IC	≥5%	1/1 (100%)	8/12 (66%)	7/21 (33%)	7/41 (17%)	2/16 (13%)	0/1 (0%)	25/92 (27%)
		CPS	≥1	1/1 (100%)	12/12 (100%)	15/21 (71%)	10/41 (24%)	10/16 (63%)	2/3 (67%)	75/94 (80%)
15	Higashino et al., 2020 [66]	TC	≥1%	6/10 (60%)	0/1 (0%)	10/33 (30%)	1/17 (6%)	5/19 (26%)	14/34 (41%)	36/127 (28%)
16	Chatzopoulos et al., 2020 [67]	CPS	≥1	13/32 (41%)						13/32 (41%)
17	Guazzo et al., 2021 [68]	CPS	≥1	5/14 (36%)		2/11 (18%)	0/17 (0%)	0/6 (0%)		7/48 (15%)
18	Sato et al., 2021 [69]	TC	≥1%	5/8 (63%)	2/4 (50%)	8/20 (40%)	2/13 (15%)	2/7 (29%)	5/10 (50%)	24/62 (39%)
		IC	≥1%	7/8 (88%)	4/4 (100%)	12/20 (60%)	2/13 (15%)	6/7 (86%)		35/62 (56%)
		CPS	≥1	7/8 (88%)	4/4 (100%)	12/20 (60%)	3/13 (20%)	6/7 (86%)		37/62 (60%)
19	Schvartsman et al., 2021 [70]	TC	≥1%	9/17 (53%)						9/17 (53%)
20	Dou et al., 2021 [41]	TC	≥1%				17/62 (27%)			17/62 (27%)
21	Chen et al., 2021 [40]	TC	≥1%				0/16 (0%)			0/16 (0%)
22	Fang et al., 2021 [71]	TC	≥1%	14/33 (42%)	8/15 (53%)	19/47 (40%)				41/95 (43%)
		IC	≥1%	13/33 (39%)	7/15 (47%)	21/47(45%)				41/95 (43%)
		CPS	≥1	25/33 (76%)	12/15 (80%)	38/47 (81%)				75/95 (79%)
23	Vos et al., 2023 [72]	TC	≥1%	1/6 (33%)		0/2 (0%)	2/25 (8%)	2/4 (50%)	2/10 (20%)	7/47 (15%)
24	Hirai et al., 2023 [28]	TC	≥1%	32/175 (18%)						32/175 (18%)
		CPS	≥1	93/175 (53%)						93/175 (53%)
25	Michaelides et al., 2023 [39]	CPS	≥5				0/12 (0%)			0/12 (0%)
26	Zerdan et al., 2023 [73]	TC	≥1%			31/118 (26%)				31/118 (26%)

<Scoring methods of PD-L1> TC: the percentage of PD-L1 positive tumor cells. IC: the percentage of PD-L1 positive immune cells. CPS: the total number of positive cells, including tumor cells and surrounding immune cells (lymphocytes and macrophages), divided by the total number of tumor cells, multiplied by 100. H-score: multipling the percentage of cells with 1+, 2+ or 3+ staining by the percentage of positive cells. ACC: acinic cell carcinoma, AdCC: adenoid cystic carcinoma, CPS: combined positive score, IC: immune cells, MEC: mucoepidermoid carcinoma, NOS: not otherwise specified, PD-L1: programmed death ligand-1, SDC: salivary duct carcinoma, SGC: salivary gland cancer, TC: tumor cells.

**Table 2 cancers-16-01205-t002:** Clinical efficacy of immune checkpoint inhibitors in patients with salivary gland cancers.

No.	Author	Number of Cases	Types of SGCs	Treatment Regimens	ORR	CBR	Median OS (M)	Median PFS (M)
1	Fayette et al., 2019 [78]	98	Total	Nivolumab	4.6%	53.2%	21.1	-
		46	AdCC		8.7%	33.3%	18.1	-
		52	Non AdCC		3.8%	14.0%	9.5	-
2	Niwa et al., 2020 [80]	24	Any	Nivolumab	4.2%	12.5%	-	-
3	Hanai et al., 2021 [82]	22	Any	Nivolumab	13.6%	-	NR	2.1
4	Ueda et al., 2022 [12]	12	Any	Nivolumab	25.0%	-	16.2	-
5	Cohen et al., 2018 [79]	26	PD-L1 positive SGC	Pembrolizumab	11.5%	23.0%	13	4.0
6	Mahmood et al., 2020 [13]	10	Any	Pembrolizumab	0%	70.0%	27.2	6.6
		10	Any	Pembrolizumab + RT	0%	50.0%	NR	4.5
7	Even et al., 2022 [83]	105	Total	Pembrolizumab	4.0%	54.1%	21.1	4.0
		28	PD-L1 positive SGC		10.7%	42.8%	-	-
		77	PD-L1 negative SGC		2.6%	58.4%	-	-
8	Sato et al., 2022 [11]	12	Any	Nivolumab or Pembrolizumab	33.3%	33.3%	13.5	4.0
9	Patel et al., 2021 [54]	26	AdCC	Nivolumab + Ipilimumab	4.0%	-	12.0	-
		35	Non AdCC		9.0%	-	NR	-
10	Vos et al., 2023 [72]	32	AdCC	Nivolumab + Ipilimumab	6.3%	34.4%	19.3	4.4
		32	Non AdCC		16.0%	18.8%	2.2	2.2
11	Rodriguez et al., 2020 [81]	25	Any	Pembrolizumab + Vorinostat	16.0%	72.0%	14.0	6.9
12	Rodriguez et al., 2023 [84]	20	Any	Nivolumab + Ipilimumab + RT	20.0%	50.0%	25.0	7.2
13	Mohamadpour et al., 2023 [85]	17	AdCC	Pembrolizumab + Lenvatinib	6.0%	82.0%	-	-

AdCC: adenoid cystic carcinoma, CBR: clinical benefit rate, ICI: Immune checkpoint inhibitor, M: months, NR: not reached, ORR: objective response rate, OS: overall survival, PD-L1: programmed death ligand-1, PFS: progression-free survival, RT: radiation therapy, SGC: salivary gland cancer.

## Data Availability

Not applicable.

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
