# Peer review of "Exploring Immunological Effects and Novel Immune Adjuvants in Immunotherapy for Salivary Gland Cancers"

_cancers, 2024, doi:10.3390/cancers16061205_

Round 1

Reviewer 1 Report

Comments and Suggestions for Authors

RE:  Exploring Immunological Effects and Novel Immune Adjuvants in Immunotherapy for Salivary Gland Cancers 

This article extensively reviews TME and its role in immunotherapy for salivary gland cancers. It provides an up-to-date overview of this topic timely.

Some points should be amended before the next round of review.

Introduction.

The authors should clarify the differential points of this review from the previous reviews

reference numbers 3 and 8.

4. The Immunological Background of Adenoid Cystic Carcinoma.

AdCC: more than 30% dying within 2 years of diagnosis

I don’t agree with this sentence. Many recurrent or metastatic AdCC survived long without any treatment because of an indolent clinical course, and there have been so many references supporting this outcome.

PSMA expression in AdCC and PSMA targeted therapy for AdCC should be presented more in detail (only PSAM-CAR-T in the text).

Minor points

Reference number #23: missing in the journal name

Figure 1 legend: Error

Reference number #27 = #30 

Line 166-167: Editing error

[END]

Comments on the Quality of English Language

Minor editing of English language

Author Response

Reviewer 1

This article extensively reviews TME and its role in immunotherapy for salivary gland cancers. It provides an up-to-date overview of this topic timely.

>We thank the reviewer for taking time to review our manuscript, and considering our manuscript interesting.

Q1. The authors should clarify the differential points of this review from the previous reviews reference numbers 3 and 8.

>Answer: In this review, we summarized the immunological TME in each histological subtype of salivary gland cancer and antigen-specific immunotherapy, which were not referred in the previous reviews by Theocharis et al. (Ref 3) and Mueller et al. (Ref 8). We have underscored the unique topics of our review in the Introduction section (P3 L111-117).

Q2. The Immunological Background of Adenoid Cystic Carcinoma. AdCC: more than 30% dying within 2 years of diagnosis. I don’t agree with this sentence. Many recurrent or metastatic AdCC survived long without any treatment because of an indolent clinical course, and there have been so many references supporting this outcome.

>Answer: We apologize for the misleading information. Although the survival of patients with bone metastasis was poor according to Sung et al. (PMID: 14623749), we agree that many patients with recurrent or metastatic AdCC especially with pulmonary metastasis survive long due to an indolent clinical course (PMID: 35662026). We have rephrased the sentence in the revised manuscript (P6 L207-210).

Q3. PSMA expression in AdCC and PSMA targeted therapy for AdCC should be presented more in detail (only PSMA-CAR-T in the text).

>Answer: We thank the reviewer for the valuable suggestion. The studies of immunohistochemistry and PSMA/positron emission tomography revealed that the positivity rates of PSMA in AdCC were 94% and 93%, respectively (PMID: 32503460 and 32089741). The study of radioligand therapy by 177Lu-PSMA-617 showed promising clinical responses (PMID: 35984529 and 34905121) suggesting that PSMA could be a targetable antigen in immunotherapy against AdCC. PSMA is a target of CAR-T cells, antibody drug conjugates, and bi-specific T cell-directed therapy in prostate cancer (PMID: 34721674). As well as prostate cancer, a phase I clinical trial of PSMA-targeting CAR-T cell therapy for patients with SGC is ongoing (NCT 04249947). We have added this information in the revised manuscript (P9 L353-355, P10 L356-361).

Q4. Reference number #23: missing in the journal name

>Answer: We have corrected the information of this reference in the revised manuscript (Ref 24 in the revised manuscript).

Q5. Figure 1 legend: Error

>Answer: We have corrected the typographical error in Figure 1 from “(M2) TAM” to “M2 TAM”.

Q6. Reference number #27 = #30 

>Answer: We have removed the duplicate reference (Ref 28 in the revised manuscript).

Q7. Line 166-167: Editing error

>Answer: We have corrected the editing error in the revised Figure 1 legend (P5 L180-181).

Reviewer 2 Report

Comments and Suggestions for Authors

Review on Sato et al”s “Exploring Immunological Effects and Novel immune Adjuvants in Immunotherapy for salivary Gland Cancers”

Dear Authors,

1.       The topic is useful, and the study seems to cover the area of possible immunotherapies in case of salivary gland cancers. The content seems fine, but the form of the study which seems more to be a scoping review, it must be addressed by the authors.

2.       The review does not follow the formal requirements of this type of review. Please, go over the checklist of PRISMA for reviews (systematic and scoping), mainly focusing on the methodology part which is fully missing from this study.

Author Response

Q1. The topic is useful, and the study seems to cover the area of possible immunotherapies in case of salivary gland cancers. The content seems fine, but the form of the study which seems more to be a scoping review, it must be addressed by the authors.

>Answer: We thank the reviewer for taking time to review our manuscript and considered it useful. We have thoroughly reviewed the existing literatures to reveal the up-to-date knowledge regarding immunotherapies in salivary gland cancer. In the rare type of cancer such as salivary gland cancer, it is difficult to address specific questions with evidence by systematic review due to the lack of sufficient data. Thus, we consider our manuscript as a scoping review instead of systematic review.  We have addressed this issue and added the research strategy in the revised manuscript (P3 L113, P3 L118-P4 L129).

Q2. The review does not follow the formal requirements of this type of review. Please, go over the checklist of PRISMA for reviews (systematic and scoping), mainly focusing on the methodology part which is fully missing from this study.

>Answer: We thank the reviewer for pointing out this important issue. We have added the Methods section including the research strategy according to the checklist of PRISMA (P3 L118-P4 L129).

Round 2

Reviewer 1 Report

Comments and Suggestions for Authors

Thank you, the authors, for correcting and revising the manuscript.

Now, I'm satisfied with this revised article.

Reviewer 2 Report

Comments and Suggestions for Authors

The study can be accepted in the present form.